# Predictive evolutionary modelling for influenza virus by site-based dynamics of mutations

Jingzhi Lou[1,2,14], Weiwen Liang [ID][3,14], Lirong Cao[1,4,14], Inchi Hu[5], Shi Zhao [ID][1,6], Zigui Chen [ID][7], Renee Wan Yi Chan [ID][8,9], Peter Pak Hang Cheung [ID][10], Hong Zheng[1], Caiqi Liu[1], Qi Li [ID][1], Marc Ka Chun Chong [ID][1,4], Yexian Zhang[2,4], Eng-kiong Yeoh[1,11], Paul Kay-Sheung Chan [ID][7,12], Benny Chung Ying Zee [ID][1,4], Chris Ka Pun Mok [ID][1,13] ✉ & Maggie Haitian Wang [ID][1,4] ✉

Influenza virus continuously evolves to escape human adaptive immunity and generates seasonal epidemics. Therefore, influenza vaccine strains need to be updated annually for the upcoming flu season to ensure vaccine effectiveness. We develop a computational approach, beth-1, to forecast virus evolution and select representative virus for influenza vaccine. The method involves modelling site-wise mutation fitness. Informed by virus genome and population sero-positivity, we calibrate transition time of mutations and project the fitness landscape to future time, based on which beth-1 selects the optimal vaccine strain. In season-to-season prediction in historical data for the influenza A pH1N1 and H3N2 viruses, beth-1 demonstrates superior genetic matching compared to existing approaches. In prospective validations, the model shows superior or non-inferior genetic matching and neutralization against circulating virus in mice immunization experiments compared to the current vaccine. The method offers a promising and ready-to-use tool to facilitate vaccine strain selection for the influenza virus through capturing heterogeneous evolutionary dynamics over genome space-time and linking molecular variants to population immune response.

The major driver of recurrent influenza epidemics is fast virus evolution that enables the influenza virus to escape from human immunity acquired from prior vaccination or infection[1]. In response, influenza vaccines need to be updated annually to match the circulating virus population[2]. Prediction of virus evolution has a critical role in ensuring the protective effect of influenza vaccines, which aids the selection of candidate vaccine strains nearly a year ahead before the arrival of next epidemic season[2,3]. We present a computational approach to predict influenza virus evolution through modeling the dynamic process of mutation adaptation at individual sites and locating the optimal wild-

[1]JC School of Public Health and Primary Care (JCSPHPC), The Chinese University of Hong Kong (CUHK), Hong Kong SAR, China. [2]Beth Bioinformatics Co. Ltd, Hong Kong SAR, China. [3]HKU-Pasteur Research Pole, School of Public Health, Li Ka Shing Faculty of Medicine, The University of Hong Kong, Hong Kong SAR, China. [4]CUHK Shenzhen Research Institute, Shenzhen, China. [5]Department of Statistics, George Mason University, Fairfax, VA, USA. [6]School of Public Health, Tianjin Medical University, Tianjin, China. [7]Department of Microbiology, CUHK, Hong Kong SAR, China. [8]Department of Paediatrics, CUHK, Hong Kong SAR, China. [9]Hong Kong Hub of Paediatric Excellence, CUHK, Hong Kong SAR, China. [10]Department of Chemical Pathology, CUHK, Hong Kong SAR, China. [11]Centre for Health Systems and Policy Research, CUHK, Hong Kong SAR, China. [12]Stanley Ho Centre for Emerging Infectious Diseases, CUHK, Hong Kong SAR, China. [13]Li Ka Shing Institute of Health Sciences, Faculty of Medicine, CUHK, Hong Kong SAR, China. [14]These authors contributed equally: Jingzhi Lou, Weiwen Liang, Lirong Cao. ✉e-mail: kapunmok@cuhk.edu.hk; maggiew@cuhk.edu.hk

type strains by a combined evaluation of multiple gene segments for considerations of vaccine strains.

Virus evolution is shaped by a complex interplay of genetic mutations, host immune response, and epidemiology[1,4]. Thus, although mutation is stochastic[5,6], the evolutionary process could be traced with the observed viral genetic and antigenic profiling in the host population[6]. The influenza virus evolves through antigenic drift on the two surface proteins, hemagglutinin (HA) and neuraminidase (NA), the primary immuno-active components of influenza vaccines. Previous studies showed that mutations at epitope sites of the HA played a dominant role in characterizing virus antigenic change and were under higher immune selection pressure[5,7]. In addition to the major antigenic substitutions, the virus evolution is also critically influenced by epistatic mutations or mutation interference effects[8–10], making predictions challenging as the evolutionary dynamics are non-uniform across genomic regions and time.

Currently, influenza vaccine strain determination involves extensive surveillance and characterization of the virus in terms of genetic, antigenic evolution, and epidemiological profiles, a global effort coordinated by the World Health Organization (WHO)[11]. While antigenic evolution of influenza virus can be mapped by the antigenic cartography based on hemagglutinin inhibition (HAI) assay data[12], genetic evolution is mostly delineated with the phylogenetic trees[13]. Current prediction methods of virus evolution mainly focus on modeling the fitness of tree parts[14–16], using sequences of the HA or HA1 segment, with epitope or antigenic data incorporated as component information[14,15,17]. For instance, Łuksza and Lässig calculated frequency of tree-clades and predicted the future predominant clade by an exponential function (the Malthusian model) with a certain growth rate[14]. Neher et al. used the local branching index (LBI) to rank tree-nodes in a given phylogeny to identify lineage with the highest fitness as a progenitor of strains in an upcoming influenza season, where fitness is estimated by integrating exponentially discounted branch-length surrounding a node[16]. Steinbrück and McHardy used allele

dynamics plots to identify the top three alleles characterizing antigenic novelty of tree-clades in a given season, pre-screened based on the epitope information, HI data, and phylogeny[17,18]. Also based on the Malthusian model, Huddleston et al. regarded strain rather than clade as a primary unit of analysis, and investigated different combination of factors to calculate growth rate for strains[15]. These methods projected the most likely predominant future lineages by tracing the fitness of clusters of strains. However, as virus evolution is driven by major antigenic substitutions, a substantial amount of information is contained in the dynamic process of the fitness of mutations, characterizing which might provide critical information for predicting genetic evolution.

## Results

### A site-based dynamic model for mutation forecasting

Our prediction method is primarily based on modeling the time-resolved frequency pattern of mutations for individual sites across virus genome segments (Fig. 1). The selective advantage of a mutation could be reflected in their growing prevalence in the host population. To make a projection into the future, the velocity of mutation frequency growth can be estimated by solving the first-order derivative of a frequency function over a period of mutation adaption in the host population, which is captured by the mutation transition time (Methods). Since the transition time is specific to substituted residues, the dynamic model is site-based and time-dependent. Scanning over the genome restores a global picture of mutation-selection. Such design shares the advantage of the agent-based model that agglomerates individual agents carrying simple rules and avoids fitting a complicated system with copious parameters or unrealistic assumptions[19]. The potential relationship between the genome-wide dynamics of mutation prevalence and population epidemics is illustrated in Supplementary Fig. 1. The site-based model is also highly computationally efficient in analyzing large genomic datasets.

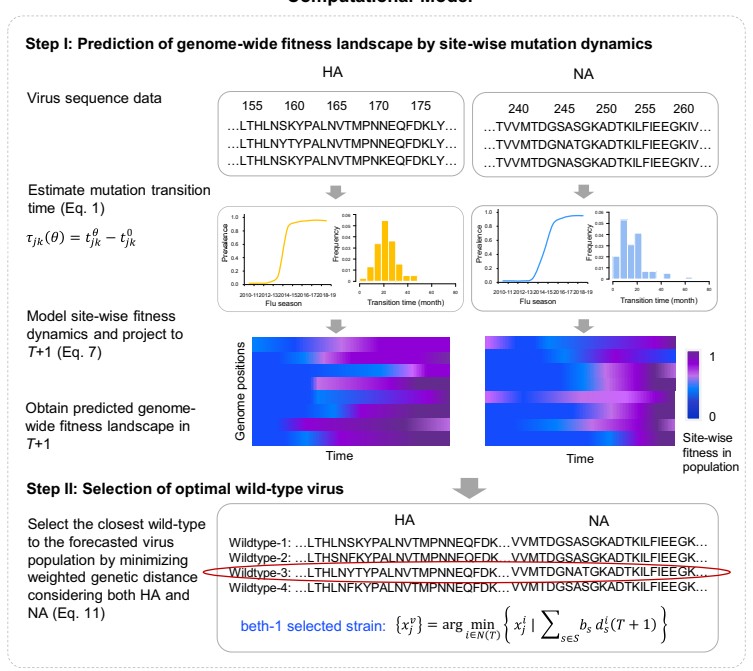
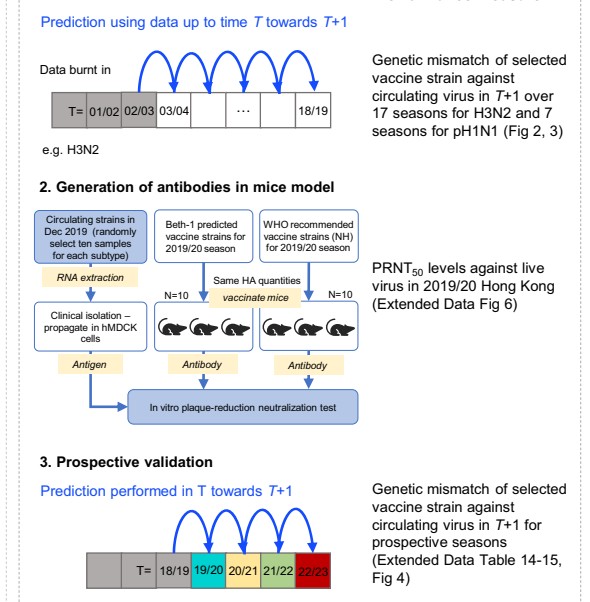

**Fig. 1 | Overview of the computational model and study design.** Site-wise fitness dynamics are modeled and projected to the next season ($T+1$). Based on the predicted genome-wide fitness landscape of future virus population, an optimal wild-type virus can be selected integrating evolutionary information of both hemagglutinin and neuraminidase genes. The prediction is validated by genetic mismatch against observed circulating viruses in the Northern Hemisphere retroactively and prospectively for pH1N1 and H3N2. In addition, mouse model is used to evaluate the antibodies elicited by predicted vaccine strain in neutralizing the clinical isolates in 2019/20 season.

Specifically, transition time in this study is defined as the duration for a mutation to emerge until it reaches an influential frequency in the population. This quantity is related to but different from the conventional fixation time, which covers a period from the first observation time of a mutation to the point when it reaches 0.99 prevalence in the population. The fixation time was reported to span over a long period of 4-32 years for the influenza virus A(H3N2)[8]. In contrast, the transition time we identified for the same viral subtype had a median length of only ~17 months and ranged between 0-7 years (Supplementary Fig. 2). As the transition time calibrates the initial period of mutation adaptation, it may have the advantage of informing the emerging genetic variants on a short-term time horizon. The transition time is determined with a frequency threshold ($\theta$) indicating fitness strength, at which the overall mutation activities are detected to influence population epidemics significantly (Supplementary Fig. 1); and it can be estimated using the virus epidemic-genetic association model we previously developed[20] (Methods).

This site-based mutation dynamic model enables the prediction for fitness of competing residues at individual sites, thereby the construction of a genome-wide fitness landscape of the virus population in future time (Fig. 1).

### Identification of optimal wild-type virus

Since regulations for influenza vaccines requires the use of a wild-type virus as vaccine strain, we next select the optimal wild-type strain based on the predicted virus population (Fig. 1). First, a consensus strain can be shaped containing all mutations showing selective advantage relative to their precedent or competing alleles in the upcoming epidemic season. Next, the optimal wild-type virus can be located by minimizing the weighted genetic distance between a candidate strain and the projected future consensus strain considering one or more proteins contained in vaccine antigen (Methods). Although only HA concentration is standardized in the current vaccine production[21], both HA and NA genes are major components of the influenza vaccines; an integrative evaluation would provide a useful tool for strain selection and evaluation.

This two-step evolution prediction and wild-type virus selection method are referred to as the "beth-1" for easy reference.

### Genetic matching of retroactively predicted vaccine strains

We applied the beth-1 to predict vaccine strains of the influenza A (H1N1)pdm09 (pH1N1) and A (H3N2) viruses. Data was collected from the Global Initiative on Sharing All Influenza Data (GISAID)[22] between 1999/2000 and 2022/23, involving a total number of 13,192 HA and 11,260 NA sequences of pH1N1, and 37,093 HA and 34,037 NA sequences of H3N2 from ten geographical regions in the Northern Hemisphere, covering North America (New York State, California State, Canada), Europe (United Kingdom, Germany, France), and Asia (Hong Kong SAR, South China provinces, Japan, Singapore) (Supplementary Table 2). Three-year data was burnt-in for model building. The prediction was performed using data up to February in season $T$, targeting the subsequent epidemic season $T+1$ (October-April next year). We calculate the average amino acid (AA) mismatch between the predicted strain and sequences of circulating viruses in the target season to determine prediction accuracy. To fully understand the performance of beth-1 in the context of existing methods, we calculated the following comparison groups: (1) WHO-recommended vaccine strains (the "current-system") for season $T+1$; (2) The local branching index (LBI) method[16], as the representative approach based on phylogenetic trees; (3) beth-1 (single protein): the beth-1 predicted strains for season $T+1$ by a single protein; (4) beth-1 (two-protein): predicted strains for season $T+1$ integrating two proteins; (5) the "answer": the observed representative strains in season $T+1$ (Methods, Supplementary Tables 3, 4).

In the retroactive data, prediction was conducted for seven seasons from 2012/13 to 2018/19 for pH1N1 and 17 seasons from 2002/03 to 2018/19 for the H3N2. beth-1 demonstrated significantly improved genetic matching to the future virus population compared to the LBI and the current-system on full-length HA and NA gene, their epitopes, and for both pH1N1 and H3N2 subtypes (Fig. 2, Supplementary Table 5). For example, the beth-1 (HA) model results in 7.5 AAs (SD 2.2) mismatch on the full-length HA protein of H3N2, while mismatch by the LBI and current-system are 9.5 AAs (SD 4.7) and 11.7 AAs (SD 5.1), respectively (pair-wise $t$-test $p$-value < 0.001) (Supplementary Table 5). The beth-1 (NA) gives 3.9 AAs (SD 1.5) mismatch on full-length NA protein of pH1N1, significantly lower than the 6.4 AAs (SD 2.1) by the LBI and 11.6 AAs (SD 4.4) by the current-system. Using the beth-1 (two-protein) model, the mismatch on the HA epitopes is 1.2 AAs (0.6) for pH1N1 and 5.1 AAs (SD 1.7) for H3N2. Particularly, the mismatch of beth-1 (two-protein) on the NA epitopes are 0.5 AA (SD 0.4) for pH1N1 and 0.6 AAs (SD 0.5) for the H3N2, close to the best possible outcome by the answer strains. In all these results, beth-1 delivers prevailingly smaller uncertainties (standard deviation) in prediction accuracy compared to the current-system and the LBI (Supplementary Table 5).

Seasonal and geographical breakdown of prediction accuracies were also analyzed. Year-by-year comparison with the LBI and current-system showed that the beth-1 gave vastly lower genetic mismatch throughout the epidemic seasons (Fig. 3, Supplementary Fig. 3, Supplementary Table 6–9). A separate analysis by geographical regions showed no systematic difference in vaccine mismatch across continents by all prediction methods and for the two influenza A subtypes (Supplementary Fig. 4, Supplementary Table 10–13).

We performed two predictions for the 2019/20 epidemic season: a prospective prediction conducted in October 2019 using training data up to June 2019; and a retroactive prediction conducted in March 2022 using data up to March 2019 to better match the current-system's timeline (Supplementary Table 14). Genetic mismatch was calculated for all prediction experiments. Between the 2019-03 and 2019-06 predictions by beth-1, the latter one recorded 0.1 less HA epitope mismatch for pH1N1 and 1.1 less HA epitope mismatch for H3N2. Both predictions showed significantly better genetic matching compared to the current-system on all protein segments of pH1N1, the NA of H3N2, and non-inferior matching in other segments. We further evaluated the immunogenicity of predicted strains for this particular season using clinical samples collected in Hong Kong (Methods, Supplementary Fig. 5). Against the viral isolates of H1 and H3 positive samples, beth-1 strains induced significantly higher neutralizing antibodies in $PRNT_{50}$ for pH1N1 and non-inferior antibodies for H3N2, compared to the current vaccines (Supplementary Fig. 6).

### Prospective predictions and validations in 2020/21–2022/23

To further validate our prediction method, we conducted prospective predictions from 2020/21 to 2022/23. The predictions were sent to the WHO before the vaccine composition meetings for the Northern Hemisphere each year (Supplementary Table 3). To fully understand the relationship between the beth-1 strains and the current vaccine viruses, we mapped them on phylogenetic trees (Fig. 4). Generally, the beth-1 predicted strains extended more into the future clusters for both influenza subtypes in all seasons of prospective validations compared to the current vaccine strains. On the epitopes of HA and NA of pH1N1 and H3N2, beth-1 demonstrated non-inferior or significantly increased genetic matching to the future circulating viruses (Supplementary Table 15). For instance, in 2020/21, beth-1 gave 2.7 AAs (SD 1.1) mismatch on the HA epitopes of pH1N1 and the current-system gave 3.8 AAs (SD 1.2) (Supplementary Table 15). In the 2021/22 season, beth-1 resulted in 4.2 AAs (SD 1.6) mismatch on the HA epitopes of H3N2, 6.6 AAs more accurate than the current-system's 10.8 AAs (SD 1.9) mismatch ($p$-value < 0.001). In the 2022/23 season, beth-1 prediction

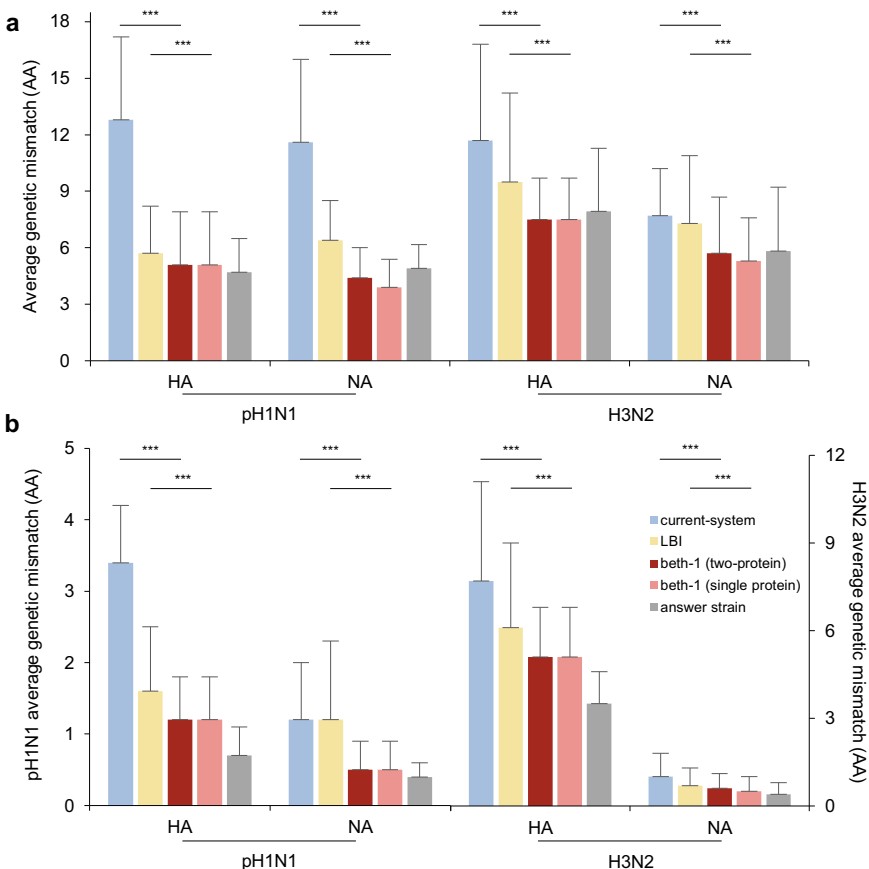

**Fig. 2 | Prediction performance in the retrospective data for influenza pH1N1 and H3N2. a** Full-length protein. **b** Epitope. The bar-plots display average genetic mismatch (amino acids, AA) over geographical regions and seven seasons in 2012/13-2018/19 for pH1N1 and 17 seasons in 2002/03-2018/19 for H3N2 ($n = 70$ region × season strata for each HA and NA of pH1N1, $n = 144$ and $n = 146$ available strata for HA and NA of H3N2, respectively). Panel (**b**) left Y-axis: average genetic mismatch of pH1N1; right Y-axis: average genetic mismatch of H3N2. Prediction methods being compared include: the current-system, LBI, beth-1 (two-protein), beth-1 (single protein), and the actual representative strain. Two-sided *p*-value is calculated using paired *t*-test on log mismatch between two methods matched by region and season. The *p*-value of current-system versus beth-1(two-protein) are 1.4e−15, 6.9e−16, <2.2e−16, <2.2e−16, respectively (**a**), and <2.2e−16, 4.9e−9, <2.2e−16, 9.0e−14, respectively (**b**). The *p*-value of LBI versus beth-1(single protein) are 6.7e−10, 1.5e−12, 1.7e−6, 7.9e−5, respectively (**a**), and 9.3e−6, 1.1e−8, 6.7e−5, 4.8e−4, respectively (**b**). In the retrospective validations, beth-1 shows significantly lower genetic mismatch on all the protein segments evaluated for the two influenza subtypes, compared to the LBI and the current-system. Error bar: standard deviation of the average genetic mismatch by region and season. ***: *p*-value < 0.001.

resulted in 1.0 AA (SD 0.7) mismatch on the HA epitopes of pH1N1 and the current-system gave 3.0 AAs (SD 0.7) mismatch (*p*-value < 0.001).

## Discussion

### Predictability of the pH1N1 and H3N2 viruses

Predictability for the two influenza subtypes pH1N1 and H3N2 can be compared on genetic mismatch of HA epitopes in the retrospective data. The ceiling of prediction can be approximated by the answer strain, that is, the observed representative strain in respective seasons. For the pH1N1 virus, the lowest possible mismatch of HA epitope was 0.7 AAs (SD 0.4) achieved by the answer strain, and the beth-1's mismatch was 1.2 AAs (SD 0.6) (Supplementary Table 5). This suggests that a highly precise hit by predictive modeling is achievable for the pH1N1, such that the resulting vaccine strain could provide an excellent match to the circulating viruses in regions of the Northern Hemisphere. However, for the H3N2 virus, the lowest possible HA epitope mismatch was as large as 3.5 AAs (SD 1.1), that is 2.8 more epitope mismatch compared to the pH1N1. Although the 5.1 AAs (SD 1.7) mismatch by beth-1 was only 1.6 residue away from the answer, the prediction for H3N2 would be ultimately bounded by the ceiling of genetic matching. The large genetic mismatch of the H3N2 answer strain might be attributed to the high genetic diversity of this virus[23,24], making the selection of a single representative wild-type strain challenging. New vaccinology strategies, such as developing broadly reactive vaccines,

designing antigens containing multiple H3N2 strains, or preparing region-specific vaccines, may provide solutions from other dimensions to enhance vaccine protection against this subtype.

### Advantages of site-based dynamic model for evolution prediction

Strain-based dynamic model often requires the estimation of a single-valued growth rate for a given genome to project future fitness, whereas a site-based dynamic model is not constrained by a constant growth rate over genome space-time. This property leads to the following three advantages in predicting virus evolution with the site-based angle. First, key mutations distributed over multiple clades could be captured as they arise, while the virus is trialing various epistatic combinations before shaping a stable lineage. Second, a site-based model avoids making assumption for directional mutation effects on fitness, which is self-evident in mutation frequency. In contrast, strain-based models often need to assume negative effect for the non-epitope mutations, to offset the genetic distance obtained from gene-based analysis in modeling evolutionary pathway[14–16], although studies suggested that the alternative might be true[25,26]. Third, the site-based dynamic model is adaptive to the altered residue fitness from epistasis and environmental factors[10,27], by sampling and re-estimating the dynamic function at each time period. When such adaptive framework is not in place, one study showed that the fitness

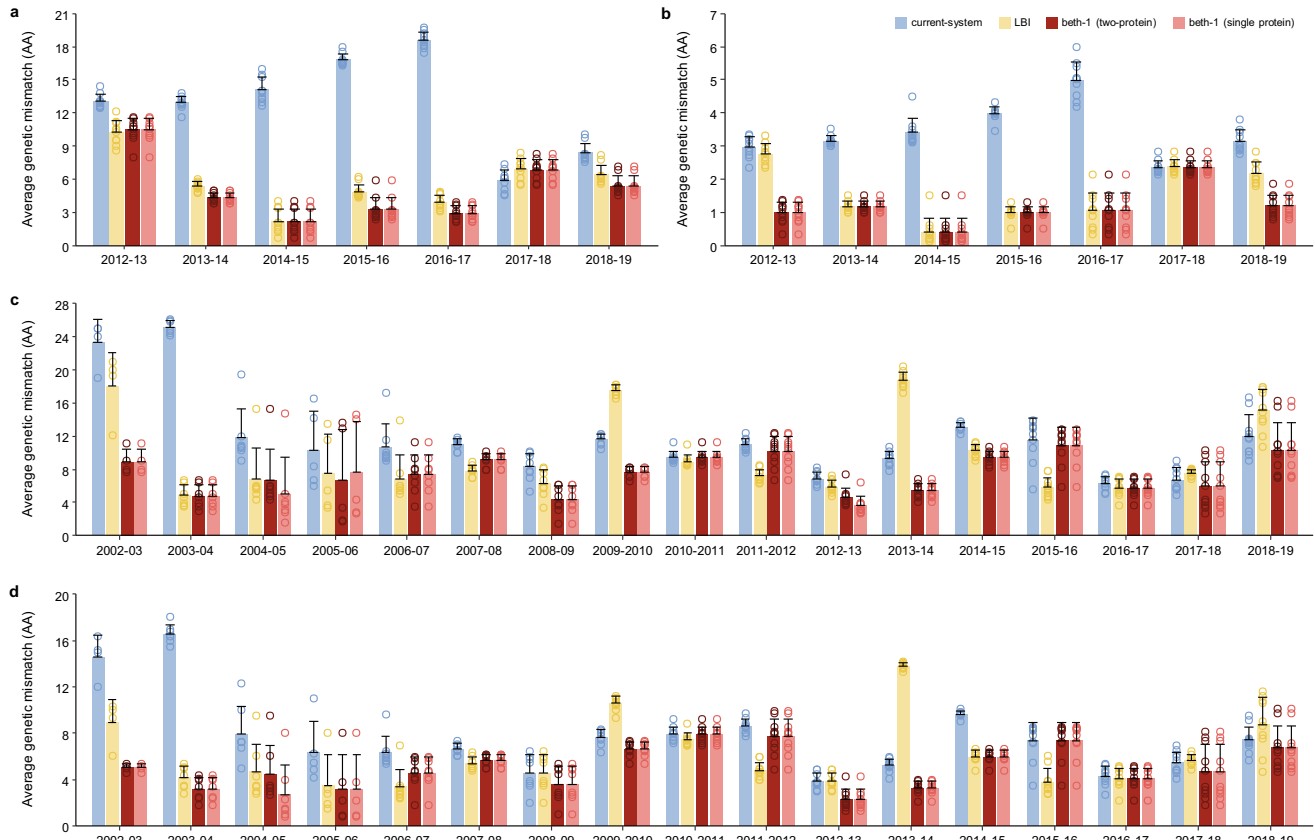

**Fig. 3 | Prediction accuracy of alternative methods for HA evolution in retrospective data by season. a** pH1N1, HA full sequence (566 codons). **b** pH1N1, HA epitopes (50 sites). **c** H3N2, HA full sequence (566 codons). **d** H3N2, HA epitopes (131 sites). Prediction accuracy is assessed by the average amino acids mismatch (Y-axis) of the predicted strains against circulating viruses in ten geographical regions in the respective epidemic seasons (X-axis). Error bar: standard deviation of the average genetic mismatch by region in a given season. The average genetic mismatch of beth-1 is prevailingly lower compared to the LBI and the current-system.

advantage estimated in the initial stage of mutation emergence cannot predict their ultimate fixation[28]. However, the proposed site-based dynamic model avoids making assumptions on the constant effect of mutations. Through adaptive estimation of fitness by genomic site and time, it projects a probable fitness landscape including all the trackable advantageous mutations to the near future.

### "Representative" viral strains by the consensus sequence in the perspective of site-based analysis

Under the perspective of strain-based analysis, a representative virus is naturally indicated by the majority vote of strain or clades. While under the perspective of site-based analysis, a genome-wide fitness landscape can be constructed for a virus population, without missing a single mutation showing selective advantage. Based on this fitness landscape, a mode estimator can be operated at individual sites and generate a consensus sequence. Therefore, the consensus sequence is a natural representative strain for a virus population based on site-wise fitness. Beth-1 in its primary objective established the theoretical framework of modeling mutation dynamics site-wise and enabled forecasting for fitness landscape into future time.

### Dissecting prediction accuracy

We can better understand the power of beth-1 by dissecting its prediction accuracy. The beth-1 (single-protein) gives slightly higher mismatch compared to the consensus strain of the predicted future by beth-1 (future-consensus) (Fig. 5), since the former one corresponds to an available wild-type virus that would be an no better representation of the predicted future compared to the future-consensus. Next, we examine performances of the future-consensus and the current virus

population (current-consensus) in the retrospective data. The result shows that the future-consensus generally improves prediction of the current-consensus over genomic regions for both influenza A subtypes (Fig. 5). It should be noted that the degree of advancement is subject to the speed of virus evolution, lead time of prediction, as well as the measurement by genetic mismatch that is under influence of viral diversity. Therefore, although the amount of advancement seems moderate, the results indicate that the site-based model can robustly add to the future that we can correctly foresee. We further analyze the 46 sites that the beth-1 correctly predicts but the current-consensus does not with respect to the answer strain for the H3N2 in the 17 retrospective seasons. Among these sites, 58.7% are epitopes and 71.7% involves physiochemical trait change, which is characterized by a conversion in charge or polarity, or volume change over 20%, and 80.5% of these sites involve either an epitope or physiochemical property change.

We next analyze the proportion of newly emerged dominant mutations captured by the site-based model year to year. Using the H3N2 as an example, on the full-length HA protein, the average number of dominant mutations arise each year is 5.3 AA (SD 4.2), estimated from the 17 seasons. The beth-1 captures 2.6 AAs (49.1%) on average of the new dominant mutations in the upcoming seasons, while LBI captures 1.5 AAs (28.3%) and the current-consensus captures 0. This result reveals an interesting fact that although the current-consensus outperforms the LBI in terms of genetic mismatch towards the future virus population[28] (Fig. 5), it forecasts no evolutionary advancement in $T+1$. Rather, the prediction accuracy achieved by the current-consensus is solely contributed by capturing the center of viral cluster, which results in a smaller spread of genetic distance from a single

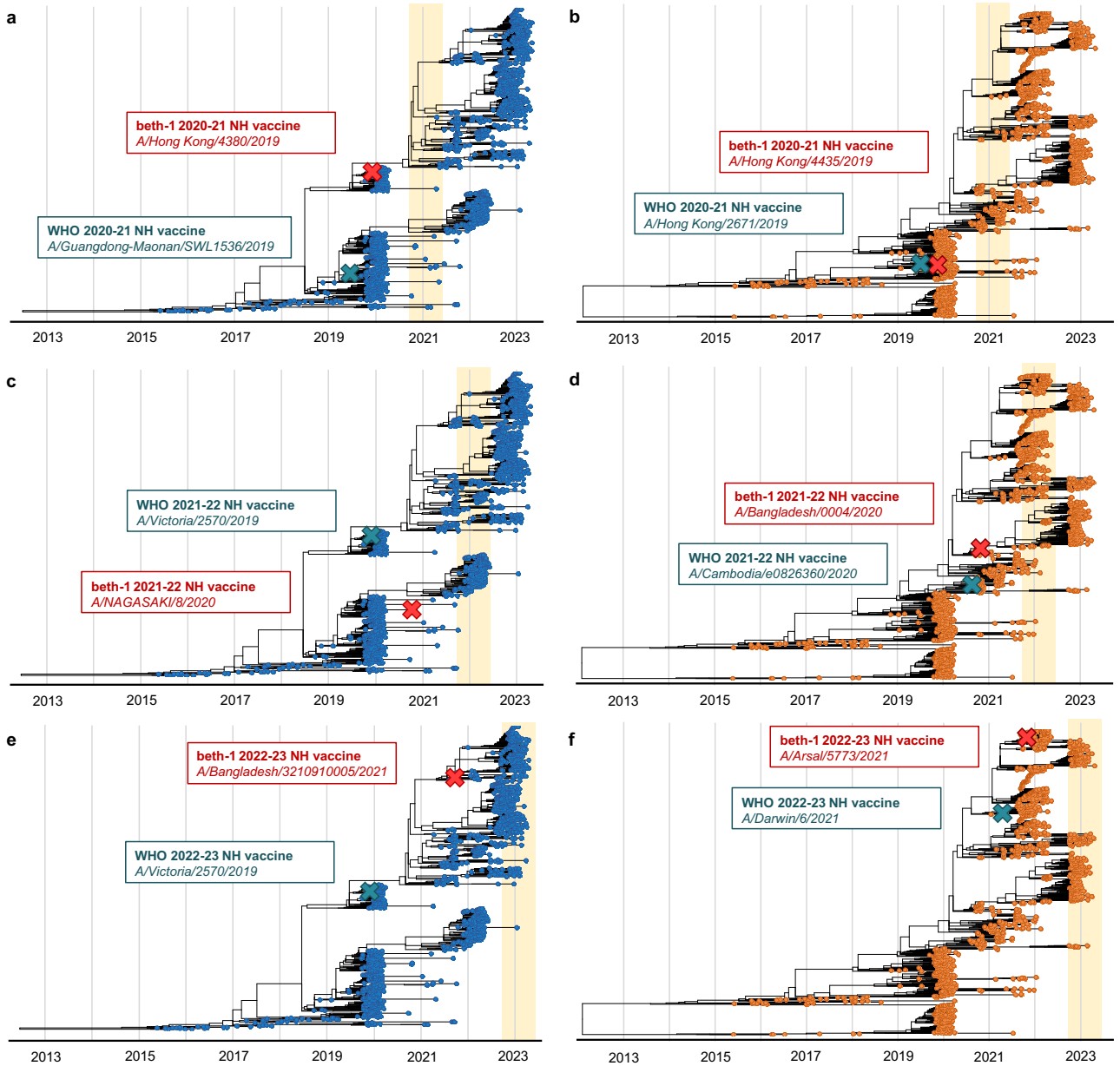

**Fig. 4 | Prospectively predicted strains on phylogenetic tree from 2020/21 to 2022/23. a** 2020/21, pH1N1. **b** 2020/21, H3N2. **c** 2021/22, pH1N1. **d** 2021/22, H3N2. **e** 2022/23, pH1N1, **f** 2023/23, H3N2. Predicted strains by beth-1 (two-protein) and the current-system are marked on phylogenetic trees with red and green, respectively. Predictions were made prospectively for the next epidemic season in the Northern Hemisphere (highlighted by yellow band in the background). The phylogeny relationship shows that beth-1's predictions are generally more advanced into the future.

strain to the circulating viruses. This also indicates that the genetic mismatch as a measure of prediction power is contributed from two aspects: the accuracy in forecasting evolutionary advancement and in locating the center of the mass of viruses. The beth-1 deals with both aspects in a simple and elegant way.

### Current WHO vaccine virus selection considerations
Currently, WHO selects vaccine strains by considering the emergence of virus with distinct genetic and antigenic characteristics, their geographical spread, and the potential loss of effective binding of antibodies from antisera of previously vaccinated subjects against the current circulating viruses[29]. These factors depict a picture of the current global virus population and their antigenic relationship to the previous representative viruses and vaccine strains, based on which recommendations of vaccine strains are made. One major

consideration of vaccine strain selection is the HAI titer for antigenic characterization[30]. Nevertheless, the HAI only evaluates one specific type of reaction, that is the prevention of HA binding to sialic acid receptor on host cells inhibited by the anti-HA head antibodies[31], while other types of immune responses are undetected, such as the antibodies against HA-stem and T-cell responses that also play important roles in protection[32,33]. Developing improved assays with broader range or finer specificity for vaccine protection may facilitate the assessment and selection of vaccine strain[29]. Another major constraint of the current vaccine strain selection process is the limited availability of high-yield virus in embryonated eggs, with which more than 95% of the current influenza vaccines are produced[34]. These candidate vaccine viruses (CVVs) are prepared from representative strains by the WHO Collaborating Centers (CC) and Essential Regulatory Labs (ERLs) before the annual WHO consultation meetings[29]. Thus, earlier

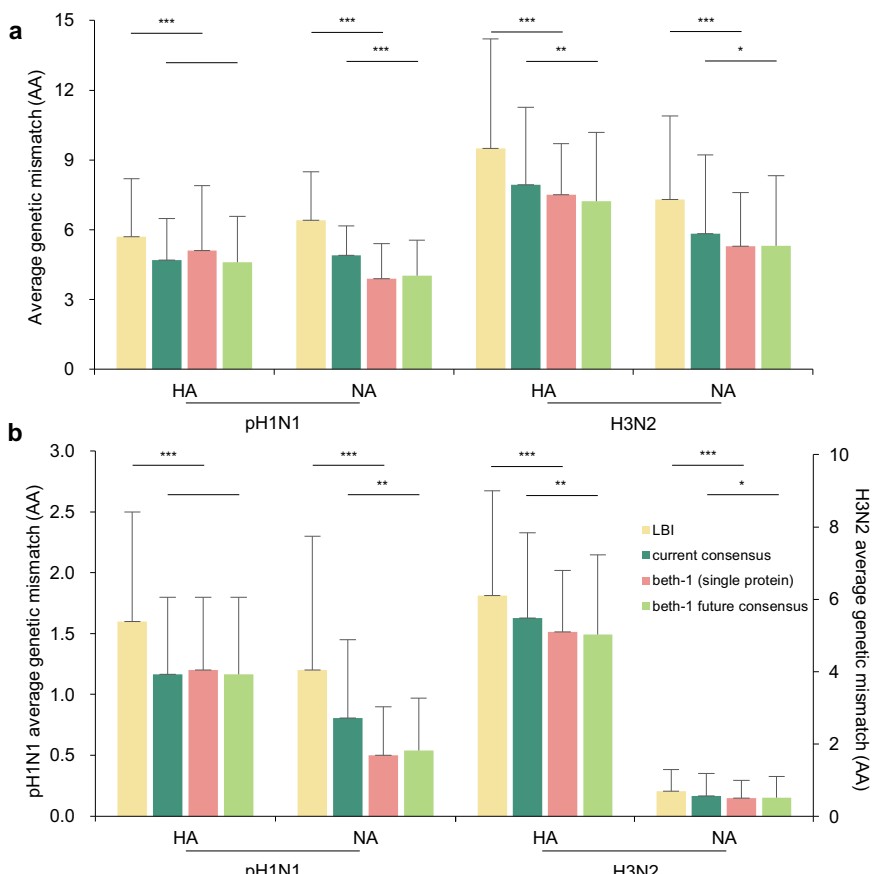

**Fig. 5 | Dissecting prediction of beth-1 by consensus strain. a** Full-length protein. **b** Epitope. The bar-plots display average genetic mismatch (amino acids, AA) over seven seasons in 2012/13-2018/19 for pH1N1 and 17 seasons in 2002/03-2018/19 for H3N2 ($n = 70$ region × season strata for each HA and NA of pH1N1, $n = 144$ and $n = 146$ available strata for HA and NA of H3N2, respectively). Panel (**b**) left Y-axis: average genetic mismatch of pH1N1; right Y-axis: average genetic mismatch of H3N2. We dissect prediction performance of the beth-1 by showing its accuracy achieved at multiple steps. The beth-1 (single protein) is compared to the future-consensus (consensus strain of the predicted future by the beth-1) and the current-consensus (consensus strain of the current virus population). Two-sided $p$-value is calculated using paired $t$-test on log mismatch between two methods matched by region and season. The $p$-value of LBI versus beth-1(single protein) are 6.7e−10, 1.5e−12, 1.7e−6, 7.9e−5, respectively (**a**), and 9.3e−6, 1.1e−8, 6.7e−5, 4.8e−4, respectively (**b**). The $p$-value of current consensus versus beth-1 future consensus are 0.061, 6.1e−5, 0.005, 0.011, respectively (**a**), and 1, 0.001, 0.009, 0.016, respectively (**b**). The future-consensus generally advances prediction of the current-consensus on the genomic segments, while beth-1 (single protein) gives slightly higher mismatch compared to the future-consensus. The LBI is displayed to replicate the previous finding involving the current-consensus strain[28]. Error bar: standard deviation of the average genetic mismatch by region and season. *: $p$-value < 0.05; **: $p$-value < 0.01; ***: $p$-value < 0.001.

prediction, even year-round projection of future representative strains may facilitate the preparation of CVVs and the subsequent vaccine strain selection, while the development of alternative vaccine technologies for influenza virus would sidestep the constraints of egg-based platform. The potential of computational optimized vaccine strains may be fully explored coupling with the availability of new production platforms, as "new wine in new wineskins".

### Limitations of the study

One major limitation in our analysis is the sparse time interval, constrained by the sequence sample size in the earlier years. In the future, this sampling gap may be gradually closed with increasing surveillance strength and global collaborations. Second, the epitope mismatch could have been underestimated as it was subjected to the known epitope sites, especially in the immune-subdominant protein segments. Nevertheless, the comparison across prediction methods was fair with the same evaluation criteria.

In summary, we have introduced a new computational method, the beth-1, for predicting influenza evolution through the site-based fitness dynamic modeling and enables strain selection considering multiple proteins. The model demonstrated promising prediction

performances in both retrospective and prospective real data applications. The framework has potentially wide applications by virology labs, vaccine manufacturers, health authorities and the WHO for indicating virus evolution and preparing vaccine virus, to facilitate influenza vaccine strain selection towards more effective vaccines.

## Methods

### Dataset

We downloaded genetic sequences of the influenza virus from the Global Initiative on Sharing All Influenza Data (GISAID)[22]. All samples were retrieved and analyzed if the strain has complete sequence and were isolated from the target epidemic seasons and geographical regions (Supplementary Table 2). The data of genetic sequences for pH1N1 spanned from 2009/10 to 2022/23, and for the H3N2 from 1999/2000 to 2022/23. Ten geographical regions in the Northern Hemisphere (NH) were considered, including North America (New York State, California State, Canada), Europe (United Kingdom, Germany, France), and Asia (Hong Kong SAR, South China provinces, Japan, Singapore). In 2020/21 and 2021/22, due to sharp decline of available samples in these regions, all sequences in NH were considered. Overall, the total number of genetic sequences used in analysis was 50,285 for

the HA and 45,297 for the NA. Sequence alignment was performed by MEGA (7.0.26)[35]. All statistical analysis were conducted in R version 4.1.3[36]. All *p*-values reported are two-sided.

**The model for evolution prediction and strain selection: beth-1**
In the following, we introduce the beth-1 model in four parts: (1) Introduction of the transition time that characterizes mutation dynamics; (2) Estimation of transition parameters; (3) Prediction of future virus fitness; (4) Identification of the wild-type virus closest to the predicted virus population. A flowchart of the method is provided in Supplementary Fig. 7.

**Transition time.** Let $x_{jk}(t)$ denote amino acid residue or nucleotide type at time $t$ and site $j, j \in J$, where $J$ is the set of sequence positions and $k \in K = [1,20]$ indexes alternative substitutions observed at site $j$. The prevalence of $x_{jk}(t)$ is denoted by $p_{jk}(t)$. We use a prevalence threshold $\theta$ to detect the mutations that demonstrate selective advantage in the host population, $\theta \in \Theta = (0,1)$. The mutation transition time describes the period for an emerging mutation to reach $\theta$ from 0 prevalence. Let $t_{jk}^0$ denote the time point when $p_{jk}(t) = 0$ and $p_{jk}(t+1) > 0$, and $t_{jk}^\theta$ denote the time when $p_{jk}(t) = \theta$ for the first time after $t_{jk}^0$. The *transition time* ($\tau$) for a particular mutation $x_{jk}(t)$ is defined as

$$\tau_{jk}(\theta) = t_{jk}^\theta - t_{jk}^0. \tag{1}$$

$\tau > 0$. Since mutations might occur multiple times in history at the same site by the prediction time $T$, we can estimate the site-specific transition time $\tau_j(\theta|T)$ using the average transition time of $x_{jk}(t)$ for $k \in K$ and $t \le T$. When no history of transition event is available at the site, its transition time is estimated by the mean transition time of mutations from the same protein in the training data (Supplementary Table 16).

**Estimation of the transition parameters.** The threshold $\theta$ is the level of prevalence for a mutation to demonstrate selective advantage in the population, which is jointly estimated with another parameter $h \ge 0$ that quantifies the duration of a mutation to remain in advantage after reaching $\theta$. With ($\theta, h$), the Effective Mutations (EMs) are those mutations that reach $\theta$ and within its effective mutation period, namely, $\tau + h$. The EMs are formally defined as the indicator function

$$m_{jk}(\theta,h,t) \triangleq I\{t_{jk}^0 \le t \le t_{jk}^\theta + h\}, \tag{2}$$

$j \in J$, $k \in K$, and $t \in [1,T]$. The overall level of mutation activities in the population can be summarized by the sum of prevalence of the EMs via the g-measure

$$g(\theta,h,t) = \mathbf{m}(\theta,h,t) \cdot \mathbf{p}(t) = \sum_{j,k} m_{jk}(\theta,h,t)p_{jk}(t), \tag{3}$$

The ($\theta, h$) is estimated through fitting the g-measure and epidemic level, $y(t)$, which equals the annual sero-positivity rate in this study. Consider the linear regression model,

$$y(t) = \beta g(\theta,h,t) + \sum_l \alpha_l z_l(t) + \varepsilon, \tag{4}$$

in which $z_l(t)$ are covariates including mean temperature, absolute humidity and season in a geographical region. The parameters $\beta$ and $\alpha_l$ are coefficients of the g-measure and the covariates, respectively, and $\varepsilon$ is a random error, $\varepsilon \sim N(0,\sigma^2)$. The ($\theta, h$) can be estimated by maximizing the goodness-of-fit of the linear regression model (Eq. 4), such that the epidemic trend is concordant with the mutation spread in the population. Suppose the R-square, $R(\theta,h)$, is used as the

goodness-of-fit statistic, we have,

$$\left(\widehat{\theta}, \widehat{h}\right) = \arg \max_{\theta \in \Theta, h \in H} R(\theta,h). \tag{5}$$

$\Theta = (0,1)$ and $H = \{0,1,2,\ldots\}$. The parameters $\theta$ and $h$ are jointly estimated for each geographical region. The fitted ($\theta, h$) can be found in Supplementary Table 16 and the estimated EMs in Supplementary Table 1. We showed in previous works that the g-measure is a good predictor for epidemic cycles of the A(H3N2)[20], A(H1N1)pdm09[37] and COVID-19[38,39], and $y(t)$ can adopt other measures for epidemic level, such as the time-varying reproduction number $R_t$[38,39].

**Prediction of future virus fitness.** Prediction of future mutation fitness can be made by solving a classical initial value problem of the differential equation,

$$\begin{cases} f'(t) = F\left[t, p_{jk}(t), \tau_j(\theta|t)\right], t > T \\ f(T) = \eta \end{cases}, \tag{6}$$

where $f(t)$ is a function describing mutation prevalence through time, $F[t, p_{jk}(t), \tau_j(\theta|t)]$ is the velocity of the prevalence change, $\eta$ is the initial value, and $T$ is the prediction time. Using the Euler's method, the projected mutation prevalence at time $T+1$ for substitution $k$ at position $j$ is,

$$\hat{p}_{jk}(T+1) = \left\{ p_{jk}(T) + F\left[T, p_{jk}(T), \tau_j\left(\hat{\theta}|T\right)\right] \right\} \cdot C, \tag{7}$$

where $C = 1/\sum_{k \in K}\{p_{jk}(T) + F[T, p_{jk}(T), \tau_j(\hat{\theta}|T)]\}$ is a normalization constant.

Equation 7 gives the forecasted future virus fitness. In practice, the time interval shall be chosen such that the training data can support a robust estimation. In the earlier seasons, only a few influenza sequence samples were available each year in many regions, thus yearly interval is chosen. Specifically,

$$F\left[T, p_{jk}(T), \tau_j\left(\hat{\theta}|T\right)\right] = \frac{\Delta p_{jk}(T)}{\Delta t} = \frac{p_{jk}(T) - p_{jk}\left[T - \tau_j\left(\hat{\theta}|T\right)\right]}{\tau_j\left(\hat{\theta}|T\right)}. \tag{8}$$

The prediction model is trained using sample sequences from the South-east Asia region that is the source of influenza A epidemics[40,41]; for the 2020/21 and 2021/22 seasons during COVID-19 pandemic, all Northern Hemisphere samples were used.

**Identification of the closest wild-type virus to the forecasted virus population.** Regulation for influenza vaccines requires the use of wild-type virus in vaccine antigen, therefore, we next identify the closest wild-type virus to the forecasted virus population. First, we use a consensus strain to represent the future virus population,

$$\left\{x_j^c(T+1)\right\} \triangleq \left\{x_{jc}(T+1)|c = k_j^*\right\}, \tag{9}$$

in which,

$$k_j^* = \arg \max_{k \in K} \hat{p}_{jk}(T+1).$$

Thus, $\{x_j^c(T+1)\}$ consists of the most advantageous mutations in $T+1$ through the mode estimator. For a given gene $s$, its future consensus strain is estimated by Eq. (9) and denoted by $\{x_{sj}^c(T+1)\}$.

When evaluating "closeness" between strains, obviously not all sites in the genome are equally important. Thus, we consider only the sites that are informative to vaccine effectiveness (VE) when

calculating genetic distance. These sites are known as the predictor codon set, which we identified previously in modeling the relationship between VE and genetic distance (VE-GD)[42,43]. The predictor codon set is composed of EMs residing in the epitope regions of the HA and NA genes[43] and is listed in Supplementary Table 1. EMs in $T+1$ are obtained using the projected $\hat{p}_{jk}(T+1)$ in Eq. (7). Denote the predictor codon set of gene $s$ identified up to $T+1$ by $W(T+1|s)$, $s \in S = \{HA, NA\}$. The genetic distance between a candidate wild-type sequence $i$ and the future consensus strain for gene $s$ is given by

$$d_s^i(T+1) = \sum_{j \in W(T+1|s)} I\{x_{sj}^i \neq x_{sj}^c(T+1)\}. \tag{10}$$

The optimal wild-type virus can be located by minimizing the weighted sum of distances on both HA and NA

$$\{x_j^v\} = \arg\min_{i \in N(T)} \left\{ x_j^i \Big| \sum_{s \in S} b_s d_s^i(T+1) \right\}, \tag{11}$$

where $N(T)$ is the set of available wild-type viruses on and before time $T$, $b_s$ are the weights for gene $s$ estimated by the VE-GD model for influenza[42,43], $b_{HA} : b_{NA} = 1 : 3$ for the pH1N1, and 1: 1.8 for the H3N2 ($p$-value < 0.001). Equation (11) gives the optimal wild-type virus identified by beth-1 for considerations of vaccine strains. For beth-1 (two-protein) model, $S = \{HA, NA\}$, and for beth-1 (single-protein) model, $S = \{HA \text{ or } NA\}$.

## Vaccine strains predicted by alternative models
The list of predicted strains by alternative methods can be found in Supplementary Table 3–4. The answer strain is set as the closest wild-type to the consensus sequence in season $T+1$ obtained using the observed data. The predicted strains of LBI[16] are obtained by running the model's source code in python (https://github.com/rneher/FitnessInference) on the same training data as the beth-1 by gene. In each season, a new tree is built for the LBI using default parameters (eps_branch = $10^{-5}$, tau = 0.0625, outgroup= "A/Puerto Rico/8/1934" for pH1N1, "A/Hong Kong/1/1968" for H3N2). When the highest LBI score corresponds to more than one strain, one of the strains is randomly picked. Two-sample $t$-test is used to compare log mismatch against circulating viruses between alternative prediction models in a given season. When comparing model performances over multiple seasons, paired $t$-test of two methods by season is used to control seasonal variation.

## Phylogenetic tree
To better visualize the predicted strains, we present them in the time-scaled maximum-likelihood phylogenies generated from TreeTime[44]. The input tree was built by IQ-TREE[45,46] with the default parameters (bootstrap analysis = ultrafast, number of bootstraps = 1000, perturbation strength = 0.5). Genetic sequences for building the tree were sampled in proportion to continent distributions in the corresponding epidemic season, targeting a total number of 400 sequences per season for each of the pH1N1 and H3N2 subtype per season. In 2020/21, all available sequences of pH1N1 in the NH were used.

## Clinical samples for animal experiment
Between October 2019 to February 2020, a total number of 117 positive H1 and 43 H3 nasopharyngeal swab samples were collected in the Prince of Wales Hospital, Hong Kong SAR. Ten samples from each of the H1 and H3 subtypes were randomly selected for virus isolation in humanized Madin-Darby canine kidney (hMDCK) cells[47]. The sequences of the HA gene from each virus was then determined by Sanger sequencing. Phylogenetic analysis showed that the viruses belonged to the same clusters of their respective subtype in these isolates, of which HA ectodomain shared >98% amino acids similarity for the pH1N1 and

>96% for H3N2. Thus, one pH1N1 and one H3N2 virus from these isolates were randomly selected for animal experiments. Human study ethics approval has been obtained from the Joint Chinese University of Hong Kong−New Territories East Cluster Clinical Research Ethics Committee.

## Cell culture
hMDCK cells and human embryonic kidney (HEK) 293 T cells were used in this study. Both cells were maintained in minimum essential medium (MEM) supplemented with 10% fetal bovine serum (FBS; Gibco), 25 mM HEPES, and 100 U/mL penicillin-streptomycin (PS; Gibco).

## Virus rescuing by reverse genetics
The H1N1 A/Puerto Rico/8/1934 (PR8) eight-plasmid reverse genetic system was used to produce the recombinant influenza virus with the predicted HA and NA genes[48]. Chimeric 6:2 recombinant viruses with six internal genes from PR8 (PB2, PB1, PA, NP, M, and NS) and 2 genes (HA and NA) from strain of interest were synthesized by Sangon Technology (Guangzhou, China) and were cloned into the pHW2000 vector[49]. The HEK 293 T cells and hMDCK cells were then mixed at a ratio of 2:1 one-day before transfection and co-cultured in a 6-well plate until they reached 70% confluence. For each recombinant virus, 16 µL TransIT®-LT1 (Mirus) and 1 µg for each of the 8 plasmids encoding the corresponding virus fragments were mixed thoroughly for transfection. The medium was replaced with 1 mL of MEM supplemented with 25 mM HEPES and 100 U/mL PS at 6-hour post transfection. At 24-hour post-transfection, another 1 mL of MEM was added, supplemented 25 mM HEPES, 100 U/mL PS, and 1 µg/mL tosylphenylalanyl chloride methyl ketone (TPCK) trypsin (Sigma). At 72-hour post-transfection, cell supernatants were inoculated into hMDCK cells maintained in MEM containing 25 mM HEPES, 100 U/mL PS, and 1 µg/mL TPCK trypsin. The viruses were harvested at 72 hours after infection and titrated by plaque assay.

## Immunization of the influenza vaccine virus in a mouse model
The research protocol of animal experiment was carried out in strict accordance with the recommendations and was approved by the Teaching and Research Committee on the Use of Live Animals (CULATR 5598-20) of the University of Hong Kong. Batches of ten 8-10 weeks old female BALB/c mice were inoculated intraperitoneally (i.p) with $10^5$ PFU of Addavax-adjuvant recombinant virus in 200 µL of phosphate-buffered saline (PBS; Gibco). At twenty-one days post immunization, peripheral blood was drawn from the immunized mice by cardiac puncture, and the serum was collected by centrifugation. All sera were treated with receptor destroying enzyme (RDE; Sigma) for 16−18 hours to reduce non-specific antibodies, then heated at 56 degrees for 30 minutes to inactivate the RDE residue. The serum was stored at −80 degrees for further experiments.

## Immunogenicity assessment based on plaque reduction neutralization test
hMDCK cells were used for virus titration by plaque assay and plaque reduction neutralization test (PRNT). hMDCK cells were prepared in a 12-well plate at least 24 hours before PRNT and cultured until obtaining a 100% confluent monolayer in each well. Two-fold serial dilutions of antiserum from 1:10 to 1:1280 were prepared, where each 100 µL was mixed with 100 PFU of viruses and incubated at 37 °C for 1 hour. Cells were washed once with PBS and inoculated with corresponding virus-antiserum mixture for further incubation at 37 °C for one hour. After that, the supernatant in each well was replaced with 3 mL of MEM supplied with a final concentration of 25 mM HEPES, 100 U/mL PS, 1% agarose gel, and 1 µg/mL TPCK trypsin. Against the live viruses, the highest dilution of the antiserum that inhibited at least 50% of the viral plaques (PRNT$_{50}$ titer) was recorded. The plate was inverted and

incubated at 37 °C for 48 hours. Finally, the cells were fixed with 4% formaldehyde and stained with 0.5% crystal violet for virus plaque counting. We recorded the highest dilution of the antiserum that inhibited at least 50% of the viral plaques as the $PRNT_{50}$ titer of the antiserum. Detailed information of the reagents and cell lines used in this study were listed in Supplementary Table 17.

## Reporting summary

Further information on research design is available in the Nature Portfolio Reporting Summary linked to this article.

## Data availability

All data used in this study is publicly available. Viral sequence data were downloaded from the global initiative on sharing all influenza data (GISAID) at http://platform.gisaid.org/ and the accession numbers were provided in the online supplementary acknowledgment table (https://github.com/mwanglab/beth-1/tree/main/acknowledgement_table).

## Code availability

The code is available at https://github.com/mwanglab/beth-1. Access and use of the code is subject to a revocable, non-transferable, and limited right for the exclusive purpose of undertaking academic or not-for-profit research. Use of the Code or any part thereof for commercial purposes requires a Commercial License Agreement from Beth Bioinformatics (info@bethbio.com).

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

## Acknowledgements

We thank the GISAID Initiative for critical surveillance efforts and open data sharing, and we greatly acknowledge the authors' contributions, submitting and originating laboratories. This work was supported by the fundings listed below. Health and Medical Research Fund, the Food and Health Bureau, the Government of the Hong Kong Special Administrative Region: INF-CUHK-1 (E.Y.), COVID190103 (M.H.W.) and 19180932 (C.K.P.M.). National Natural Science Foundation of China: 32322088 (M.H.W.) and 71974165 (M.K.C.C.). The Chinese University of Hong Kong Direct Grant: PIEF/Ph2/COVID/06 and 2022.02 (M.H.W.). Guangdong-Hong Kong-Macau Joint Laboratory of Respiratory Infectious Disease: 20191205 (C.K.P.M.). Visiting scientist scheme from Lee Kong Chian School of Medicine, Nanyang Technological University, Singapore (C.K.P.M.). Research Grants Committee, General Research Fund: 16301319 and C6036-21G (P.P.H.C.)

## Author contributions

M.H.W. conceived the study. M.H.W., J.L. and L.C. developed the prediction method. J.L. wrote code, ran the model and analyzed output data. C.K.P.M. designed the animal experiment. P.K.C. and Z.C. provided clinical samples. W.L. and C.K.P.M. conducted the immunology experiments. L.C., S.Z., H.Z., Y.Z., C.L. and Q.L. contributed to data collection and interpretation. E.Y., M.H.W., M.K.C.C. and C.K.P.M. acquired funding. M.H.W. and J.L. wrote the original manuscript. L.C., C.K.P.M. and W.L. revised the manuscript. I.H., Z.C., R.W.Y.C., P.P.H.C., and M.K.C.C. reviewed and edited the manuscript. B.C.Y.Z. and E.Y. supervised the project.

## Competing interests

The Chinese University of Hong Kong filed a pending patent WO2019242597A1, covering the method described in this manuscript, listing M.H.W., J.L., M.K.C.C. and B.C.Y.Z. as inventors. M.H.W. and B.C.Y.Z. are shareholders of Beth Bioinformatics Co., Ltd. B.C.Y.Z. is a shareholder of Health View Bioanalytics Ltd. The remaining authors declare no competing interests.
