## [Peer Review File · Nature Communications]

Predictive evolutionary modelling for influenza virus by site-based dynamics of mutationsEditorial Note: This manuscript has been previously reviewed at another journal that is not operating a transparent peer review scheme. This document only contains reviewer comments and rebuttal letters for versions considered at *Nature Communications*.

REVIEWER COMMENTS

Reviewer #3 (Remarks to the Author):

Summary

Thank you to the authors for their detailed responses to the previous reviews. These responses addressed most of my concerns except for a single major point made by myself and previous reviewer 4 which I describe below.

Major comment

In my first review of this paper, I asked the authors to compare their method to the “consensus sequence” method proposed by Barrat-Charlaix et al. 2021, since those authors had found that the consensus sequence for a current season could be as good or better a representative of the future population than a strain picked by LBI. In response, the Lou et al. created a figure (Figure R5, attached here as "Lou-et-al-rebuttal-1-figure-R5.png") showing the average genetic mismatch for the “current system”, LBI, current consensus, beth-1 future consensus, and the “answer strain”. This figure showed two important results:

1. the current consensus sequence often produced better estimates of the future than LBI
2. the beth-1 approach often produced better estimates of the future than the consensus sequence

These results were important because they replicated the findings from Barrat-Charlaix et al. 2021 and they showed that the beth-1 method still adds value to the state of influenza forecasting methods. Since beth-1 operates by producing a consensus sequence of future strains based on site-specific effects, the current season’s consensus sequence provides a natural control and baseline for the beth-1 method by showing what one might predict the future to look like without any site-specific model of evolution.

I appreciate that the authors have attempted to address the relationship between their method and the current season consensus with a new section in the discussion (“Dissecting prediction accuracy”). However, this section introduces a new forecasting target which tests how well LBI and beth-1 capture “the new dominant mutations in the upcoming season”. This is not the forecasting target used earlier in the paper or a target used by the “current system” or LBI methods. As a result, this section ends up confusing the relationships between these different methods.

In response to reviewer 4’s question about the current consensus method, the authors reply that these kinds of methods “only summarize the current virus population”. That’s ok, though, since the “current system” and LBI do the same thing: they summarize the current population and try to pick the most representative strain of the future population. These methods don’t explicitly predict the future with a time model. In this way, beth-1 is unique, since it actually tries to predict the mutational composition of the future population. If the current system and LBI can be compared to beth-1, then the current season’s consensus can be, too. Even when beth-1’s performance is modestly better than the current consensus sequence (as reviewer 4 mentions), beth-1 allows us to better understand the source of its predictions by highlighting the mutations at specific sites that should be important in the future. This ability to connect beth-1 predictions to specific biological changes is an important feature of any forecasting system and one that the model-free current consensus approach lacks.

Instead of the new discussion section added in this latest revision, I strongly suggest that the authors include the current season consensus results in their main results figures and text. The results will strengthen their argument that beth-1 advances the field much more effectively than the new discussion section does. The authors can confidently state that, even when their model’s predictions are not a major improvement over a model-free consensus, their model can explain the source of its predictions in a way that can help WHO decision-makers interpret the predictions.

Minor comments

* Line 226: "...dynamic model is not constraint by..." should be "...dynamic model is not constrained by..."

* Line 237: "...not in place, study showed..." should be "...not in place, one study showed..."

Reviewer #3 (Remarks on code availability):

The source code was easy to read and understand. I used this code several times throughout my review to better understand technical aspects of the methods that were not clear in the manuscript and to check specific results.

Reviewer #4 (Remarks to the Author):

The manuscript by Lou et al has continued to improve throughout the review process. That said, I still have reservations about the model and its implications.

At the request of the editor, I also considered the responses to previous comments by reviewer 2. I share reviewer 2's concerns and provide comments on those responses as well.

1. As originally suggested by Reviewer 2, I think it is essential to segregate the current work from any discussion of vaccine effectiveness. In preparing this review, I read all the work referenced by the authors on the relationship between genetic distance and vaccine effectiveness. It should be noted that all that work is by the authors of the present manuscript. While I appreciate that 3 of the 4 studies referred to by the authors have been published in peer reviewed journals, none of those studies describe the methods clearly enough for me to evaluate the validity of their findings. This echoes reviewer 2's concerns about the clarity of those studies and the methods section of the present study. With this in mind, and because vaccine effectiveness is not an essential part of the present study, I strongly suggest removing all mentions of the relationship between genetic distance and vaccine effectiveness, particularly the text at lines 161-167.

2. The description of the methods in the current version of the manuscript is unnecessarily complicated. Specifically, the equations and accompanying text in the methods section are high level generalizations of much simpler concepts. For example, while equation 2 is technically correct, the implementation used by the authors is essentially just a correlation. Other sections of the methods referring to optimization criteria are just regression. As mentioned in comment 1, this was a persistent concern for Reviewer 2, and it still has not been suitably addressed in the revised manuscript. As far as I can tell the prediction model is technically sound, but the methods section should be extensively re-written to encourage reproducibility and to specify exactly what the authors did, not a mathematical generalization of it. Given that lack of clarity in methods is a problem for this and multiple other studies from the originating group, retaining the services of a statistically trained copyeditor seems essential.

3. As pointed out by reviewer 3, and by me in previous reviews, the comparison of beth-1 to other methods remains problematic. In previous versions of the study, the authors included comparisons with the current-consensus strain. These comparisons are not present in the current version of the study, but it is essential that they be included in the main text figures – particularly because they are very comparable. The authors' rationale for why they have not been included directly contradicts Barrat-Charlaix et al. By that same rationale, they should also not include WHO's approach.

4. Further comparison needs to be pursued with existing methods, particularly that of Luksza and Lassig, Nature, 2014. Previously the author's have stated that the code is not publicly available and that they have been unable to obtain the code from Luksza and Lassig. Given that the most relevant comparison for beth-1 is the Luksza and Lassig model, and that the Luksza and Lassig paper was published by NPG, surely it is possible to compel them to share the model.

5. Further analyses need to be done to determine to if beth-1 is doing something useful in the context of vaccine strain selection. As mentioned above, I suspect that beth-1 is technically sound, but it is still not clear to me that a 1-2 amino acid reduction in genetic distance is meaningful. For all considered seasons, it should be possible to directly compare the viruses identified by beth-1,

LBI, and current-consensus by aligning the amino acid sequences, identifying the sequence differences and studying the properties of those amino acid differences. Are the amino acid differences for the LBI and consensus-sequence viruses compared to the beth-1 virus likely to have an impact on antigenicity? Are they in antigenic sites? Are they associated with changes in volume, charge, or polarity? Hopefully, the answer to all of these questions is yes. I appreciate that predicting evolution is interesting all by itself, but this manuscript about is about vaccine strain selection and these questions are important for that process.

6. Further copyediting is needed to remove all text related to the animal study.

7. Currently, phylogenetic trees appear to be constructed using neighbor joining. This method is fast but highly error prone. Something like IQtree should be used instead.

Response to Reviewers

Reviewer #3

Summary

(1) *“Thank you to the authors for their detailed responses to the previous reviews. These responses addressed most of my concerns except for a single major point made by myself and previous reviewer 4 which I describe below.”*

Response: Thank you so much for your review! We also have learnt greatly from your constructive comments.

Major comment

(2) *“In my first review of this paper, I asked the authors to compare their method to the “consensus sequence” method proposed by Barrat-Charlaix et al. 2021, since those authors had found that the consensus sequence for a current season could be as good or better a representative of the future population than a strain picked by LBI. In response, the Lou et al. created a figure (Figure R5, attached here as "Lou-et-al-rebuttal-1-figure-R5.png") showing the average genetic mismatch for the “current system”, LBI, current consensus, beth-1 future consensus, and the “answer strain”. This figure showed two important results:*

- 1. the current consensus sequence often produced better estimates of the future than LBI*
- 2. the beth-1 approach often produced better estimates of the future than the consensus sequence*

These results were important because they replicated the findings from Barrat-Charlaix et al. 2021 and they showed that the beth-1 method still adds value to the state of influenza forecasting methods. Since beth-1 operates by producing a consensus sequence of future strains based on site-specific effects, the current season’s consensus sequence provides a natural control and baseline for the beth-1 method by showing what one might predict the future to look like without any site-specific model of evolution.

I appreciate that the authors have attempted to address the relationship between their method and the current season consensus with a new section in the discussion (“Dissecting prediction accuracy”). However, this section introduces a new forecasting target which tests how well LBI and beth-1 capture “the new dominant mutations in the upcoming season”. This is not the forecasting target used earlier in the paper or a target used by the “current system” or LBI methods. As a result, this section ends up confusing the relationships between these different methods.

(In response to reviewer 4’s question about the current consensus method, the authors reply that these kinds of methods “only summarize the current virus population”. That’s ok, though, since the “current

system” and LBI do the same thing: they summarize the current population and try to pick the most representative strain of the future population. These methods don’t explicitly predict the future with a time model. In this way, beth-1 is unique, since it actually tries to predict the mutational composition of the future population. If the current system and LBI can be compared to beth-1, then the current season’s consensus can be, too. Even when beth-1’s performance is modestly better than the current consensus sequence (as reviewer 4 mentions), beth-1 allows us to better understand the source of its predictions by highlighting the mutations at specific sites that should be important in the future. This ability to connect beth-1 predictions to specific biological changes is an important feature of any forecasting system and one that the model-free current consensus approach lacks.)

Instead of the new discussion section added in this latest revision, I strongly suggest that the authors include the current season consensus results in their main results figures and text. The results will strengthen their argument that beth-1 advances the field much more effectively than the new discussion section does. The authors can confidently state that, even when their model’s predictions are not a major improvement over a model-free consensus, their model can explain the source of its predictions in a way that can help WHO decision-makers interpret the predictions.”

Response: Thank you very much for your comments. We added the comparison of beth-1 future consensus, current-consensus and the LBI in the new Figure 5 (**Figure R1** below) in the manuscript.

Figure R1 (Fig. 5 in the manuscript) Dissecting prediction of beth-1 by consensus strain.

a, Full-length protein. **b**, Epitope.

The following text is added to the manuscript.

(page 9, line 242) **“Dissecting prediction accuracy.** We can better understand the power of beth-1 by dissecting its prediction accuracy. The beth-1 (single-protein) gives slightly higher mismatch compared to the consensus strain of the predicted future by beth-1 (future-consensus) (**Fig 5**), since the former one corresponds to an available wild-type virus that would be an no better representation of the predicted future compared to the future-consensus. Next, we examine performances of the future-consensus and the current virus population (current-consensus) in the retrospective data. The result shows that the future-consensus generally improves prediction of the current-consensus over genomic regions for both influenza A subtypes (**Fig 5**). It should be noted that the degree of advancement is subject to the speed of virus evolution, lead time of prediction, as well as the measurement by genetic mismatch that is under influence of viral diversity. Therefore, although the amount of advancement seems moderate, the results indicate that the site-based model can robustly add to the future that we can correctly foresee. We further analyze the 46 sites that the beth-1 correctly predicts but the current-consensus does not with respect to the answer strain for the H3N2 in the 17 retrospective seasons. Among these sites, 58.7% are epitopes and 71.7% involves physiochemical trait change, which is characterized by a conversion in charge or polarity, or volume change over 20%, and 80.5% of these sites involve either an epitope or physiochemical property change.

We next analyze the proportion of newly emerged dominant mutations captured by the site-based model year to year. Using the H3N2 as an example, on the full-length HA protein, the average number of dominant mutations arise each year is 5.3 AA (SD 4.2), estimated from the 17 seasons. The beth-1 captures 2.6 AAs (49.1%) on average of the new dominant mutations in the upcoming seasons, while LBI captures 1.5 AAs (28.3%) and the current-consensus captures 0. This result reveals an interesting fact that although the current-consensus outperforms the LBI in terms of genetic mismatch towards the future virus population²⁸ (**Fig. 5**), it forecasts no evolutionary advancement in $T+1$. Rather, the prediction accuracy achieved by the current-consensus is solely contributed by capturing the center of viral cluster, which results in a smaller spread of genetic distance from a single strain to the circulating viruses. This also indicates that the genetic mismatch as a measure of prediction power is contributed from two aspects: the accuracy in forecasting evolutionary advancement and in locating the center of the mass of viruses. The beth-1 deals with both aspects in a simple and elegant way.”

We feel that adding the comparison of beth-1 to the current-consensus in this section of “Dissecting prediction accuracy” is more appropriate than adding it to the existing Figure 2. Because this comparison naturally fits in the context – its previous paragraph revealed the new interpretation of consensus strain that it is the summary statistic of site-wise fitness and therefore a good representation of a given virus population. Next, in these two paragraphs, we build upon the consensus strain to dissect

prediction accuracy, in the perspective of analyzing the absolute advancement in AAs by the future-consensus comparing to the current-consensus and the perspective of capturing new dominant mutations. Furthermore, all strains in Figure 2 are wild-type viruses for consideration of vaccine strains, while the consensus strains may not have a wild-type correspondence. The future-consensus is also an intermediate output of beth-1 and adding it to Figure 2 could be confusing.

Minor comments

(3) * Line 226: “...dynamic model is not constraint by...” should be “...dynamic model is not constrained by...”

Response: Thank you very much. It is corrected.

(4) * Line 237: “...not in place, study showed...” should be “...not in place, one study showed...”

Response: Thank you very much. Corrected.

Reviewer #3 (Remarks on code availability):

(5) “The source code was easy to read and understand. I used this code several times throughout my review to better understand technical aspects of the methods that were not clear in the manuscript and to check specific results.”

Response: Thank you so much for your checking and comments!

Reviewer #4

“The manuscript by Lou et al has continued to improve throughout the review process. That said, I still have reservations about the model and its implications.

At the request of the editor, I also considered the responses to previous comments by reviewer 2. I share reviewer 2’s concerns and provide comments on those responses as well.”

Response: Thank you very much for your dedicated review. We have carefully revised the manuscript following your comments.

(1) *“As originally suggested by Reviewer 2, I think it is essential to segregate the current work from any discussion of vaccine effectiveness. In preparing this review, I read all the work referenced by the authors on the relationship between genetic distance and vaccine effectiveness. It should be noted that all that work is by the authors of the present manuscript. While I appreciate that 3 of the 4 studies referred to by the authors have been published in peer reviewed journals, none of those studies describe the methods clearly enough for me to evaluate the validity of their findings. This echoes reviewer 2’s concerns about the clarity of those studies and the methods section of the present study. With this in mind, and because vaccine effectiveness is not an essential part of the present study, I strongly suggest removing all mentions of the relationship between genetic distance and vaccine effectiveness, particularly the text at lines 161-167.”*

Response: Thank you for the comments. We have removed all text relating to VE estimation in the result and method section, including the original lines 161-167.

(2) *“The description of the methods in the current version of the manuscript is unnecessarily complicated. Specifically, the equations and accompanying text in the methods section are high level generalizations of much simpler concepts. For example, while equation 2 is technically correct, the implementation used by the authors is essentially just a correlation. Other sections of the methods referring to optimization criteria are just regression. As mentioned in comment 1, this was a persistent concern for Reviewer 2, and it still has not been suitably addressed in the revised manuscript. As far as I can tell the prediction model is technically sound, but the methods section should be extensively re-written to encourage reproducibility and to specify exactly what the authors did, not a mathematical generalization of it. Given that lack of clarity in methods is a problem for this and multiple other studies from the originating group, retaining the services of a statistically trained copyeditor seems essential.”*

Response: Thank you very much for your comments. We have invited Prof. Inchi Hu, PhD in Statistics (Stanford) and Fellow of Institute of Mathematical Statistics, to edit and advise on the writing of Method. The improvements made are summarized as follows:

(1) The computational method is divided into 4 parts to introduce, and a flowchart is added to illustrate the relationship between the parts (**Figure R2** below, or new **Supplementary Fig. 7**);

Figure R2 (or Supplementary Fig. 7). beth-1 method flowchart.

(2) High level generalizations are avoided as much as possible in the revised version. The regression mentioned by the reviewer is described specifically in **Eq. 4**. The general goodness-of-fit statistics $S()$ in the original Equation 2 is directly written as the R-square (correlation) in the new **Eq. 5**, as follows,

(page 13, line 358) “Suppose the R-square, $R(\theta, h)$, is used as the goodness-of-fit statistic, we have,

$$(\hat{\theta}, \hat{h}) = \arg \max_{\theta \in \Theta, h \in H} R(\theta, h). \quad \text{Eq. 5}$$

$\Theta = (0,1)$ and $H = \{0,1,2, \dots\}$.”

(3) Notations are simplified. For example, $k \in K_j$ is simplified to $k \in K = [1, 20]$ for amino acid sequences.

(4) More text is added to explain interpretation of the quantities.

We hope these revisions help to make reading of the Method easier.

(3) *“As pointed out by reviewer 3, and by me in previous reviews, the comparison of beth-1 to other methods remains problematic. In previous versions of the study, the authors included comparisons with the current-consensus strain. These comparisons are not present in the current version of the study, but it is essential that they be included in the main text figures – particularly because they are very comparable. The authors’ rationale for why they have not been included directly contradicts Barrat-Charlaix et al. By that same rationale, they should also not include WHO’s approach.”*

Response: Thank you very much for the comments. We have added the comparison to the current-consensus strain in the new **Figure 5** (or **Figure R1** in this response letter). The text accompanying the analysis is added to line 242-258, in page 9-10 of the manuscript, which is also under the response to Reviewer 3’s comment (1).

(4) *“Further comparison needs to be pursued with existing methods, particularly that of Luksza and Lassig, Nature, 2014. Previously the author’s have stated that the code is not publicly available and that they have been unable to obtain the code from Luksza and Lassig. Given that the most relevant comparison for beth-1 is the Luksza and Lassig model, and that the Luksza and Lassig paper was published by NPG, surely it is possible to compel them to share the model.”*

Response: Thank you for the comments. We are unable to obtain the code from Luksza and Lassig. However, some reference for their model can be drawn from Neher et al (2014), who showed that the performance of the LBI is comparable to Luksza and Lassig (2014)’s prediction.

(5) *“Further analyses need to be done to determine to if beth-1 is doing something useful in the context of vaccine strain selection. As mentioned above, I suspect that beth-1 is technically sound, but it is still not clear to me that a 1-2 amino acid reduction in genetic distance is meaningful. For all considered seasons, it should be possible to directly compare the viruses identified by beth-1, LBI, and current-consensus by aligning the amino acid sequences, identifying the sequence differences and studying the properties of those amino acid differences. Are the amino acid differences for the LBI and consensus-sequence viruses compared to the beth-1 virus likely to have an impact on antigenicity? Are they in antigenic sites? Are they associated with changes in volume, charge, or polarity? Hopefully, the answer to all of these questions is yes. I appreciate that predicting evolution is interesting all by itself, but this manuscript about is about vaccine strain selection and these questions are important for that process.”*

Response: Thank you very much for the comments. We compare the amino acid (AA) differences between the beth-1 strain and alternative methods in the 17 retrospective seasons using the H3N2. We summarized the proportion of sites that are epitopes or involve a change in physiochemical property, which is characterized by either converting the charge, polarity, or a volume change over 20%. The

results show that among the 46 sites on HA that beth-1 correctly captured in the 17 seasons while the current-consensus strain did not, 58.7% are epitopes, 71.7% involves physiochemical trait change, and 80.5% of the sites are either epitopes or involve a conversion of physiochemical property. Comparing to the LBI, among the 74 sites that beth-1 correctly predicts while the LBI does not, the proportions of AA involving epitope, physiochemical change, and a change in either one of the two traits are 54.1%, 66.2% and 71.6%, respectively.

Method of calculation:

The physio-chemical properties of amino acids are referenced from Biro J.C. (2006), Kyte *et al* (1982) and Zamyatnin A.A. (1972). A charge change is evaluated by the isoelectric point (pI). pI < 6: negative charge; pI > 6: positive charge; pI = 6: neutral charge. A polarity change is referenced by the hydrophathy index (h). h < 0: hydrophilic; h > 0: hydrophobic. The volume change is the relative volume difference between two amino acids.

The AA analysis of beth-1 versus the current-consensus strain is added to the manuscript,

(page 9, line 254) “We further analyze the 46 sites that the beth-1 correctly predicts but the current-consensus does not with respect to the answer strain for the H3N2 in the 17 retrospective seasons. Among these sites, 58.7% are epitopes and 71.7% involves physiochemical trait change, which is characterized by a conversion in charge or polarity, or volume change over 20%, and 80.5% of these sites involve either an epitope or physiochemical property change.”

References:

- Biro J. C. Amino acid size, charge, hydrophathy indices and matrices for protein structure analysis. *Theoretical biology & medical modelling*, 3, 15. (2006).
- Kyte, J., & Doolittle, R. F. A simple method for displaying the hydrophathic character of a protein. *Journal of molecular biology*, 157(1), 105–132. (1982).
- Zamyatnin A. A. Protein volume in solution. *Progress in biophysics and molecular biology*, 24, 107–123. (1972).

(6) “*Further copyediting is needed to remove all text related to the animal study.*”

Response: The animal study is presented objectively and only takes two sentences in the main text (line 172- 176). We felt that keeping it in the manuscript does not harm the interpretation of the study.

(7) “*Currently, phylogenetic trees appear to be constructed using neighbor joining. This method is fast but highly error prone. Something like IQtree should be used instead.*”

Response: Thank you for your comment. Following your suggestion, the IQtree is used to re-construct the phylogenetic tree, which is used as the input tree to TreeTime to obtain the time-scaled maximum-likelihood phylogenies (**Figure 4**). The new tree looks the same as the previous version. The output of

both the IQ trees and TreeTime are stored in gtihub (https://github.com/mwanglab/beth-1/tree/main/figure/phylogenetic_tree).

REVIEWERS' COMMENTS

Reviewer #4 (Remarks to the Author):

After many iterations, I feel like the manuscript is as good as it can be. I have nothing further to add to previous comments. My lone remaining comment is that the study might benefit from a more thorough abstract but I leave this to the discretion of the editor and authors.